# Development and validation of clinical prediction models for breast cancer incidence and mortality: a protocol for a dual cohort study

Ashley Kieran Clift [ID] ,[1,2] Julia Hippisley-Cox [ID] ,[2] David Dodwell [ID] ,[3] Simon Lord,[4] Mike Brady,[4] Stavros Petrou [ID] ,[2] Gary S. Collins [ID] [5]

[1]Cancer Research UK Oxford Centre, University of Oxford, Oxford, UK
[2]Nuffield Department of Primary Care Health Sciences, University of Oxford, Oxford, UK
[3]Nuffield Department of Population Health, University of Oxford, Oxford, UK
[4]Department of Oncology, University of Oxford, Oxford, UK
[5]Centre for Statistics in Medicine, University of Oxford, Oxford, UK

**Correspondence to**
Dr Ashley Kieran Clift;
ashley.clift@phc.ox.ac.uk

## ABSTRACT

**Introduction** Breast cancer is the most common cancer and the leading cause of cancer-related death in women worldwide. Risk prediction models may be useful to guide risk-reducing interventions (such as pharmacological agents) in women at increased risk or inform screening strategies for early detection methods such as screening.

**Methods and analysis** The study will use data for women aged 20–90 years between 2000 and 2020 from QResearch linked at the individual level to hospital episodes, cancer registry and death registry data. It will evaluate a set of modelling approaches to predict the risk of developing breast cancer within the next 10 years, the 'combined' risk of developing a breast cancer and then dying from it within 10 years, and the risk of breast cancer mortality within 10 years of diagnosis. Cox proportional hazards, competing risks, random survival forest, deep learning and XGBoost models will be explored. Models will be developed on the entire dataset, with 'apparent' performance reported, and internal-external cross-validation used to assess performance and geographical and temporal transportability (two 10-year time periods). Random effects meta-analysis will pool discrimination and calibration metric estimates from individual geographical units obtained from internal-external cross-validation. We will then externally validate the models in an independent dataset. Evaluation of performance heterogeneity will be conducted throughout, such as exploring performance across ethnic groups.

**Ethics and dissemination** Ethics approval was granted by the QResearch scientific committee (reference number REC 18/EM/0400: OX129). The results will be written up for submission to peer-reviewed journals.

## Strengths and limitations of this study

► The linkage of primary care, hospital, cancer registry and death registry data for millions of women will permit accurate ascertainment of cases and predictor variable values.

► The sample size (>11 million) will present the largest ever study to develop and validate risk prediction models for breast cancer.

► Internal-external cross-validation of models across >1800 practices in 10 regions and in 2 time periods will permit robust evaluation of performance, performance heterogeneity, and geographical and temporal transportability.

► An external validation in an independent dataset will be performed, which will analyse model performance according to ethnicity, geographical region and other key characteristics.

► Genetic information and radiological images are not available in the development dataset, which may be relevant for breast cancer risk prediction (although having undergone mammography is recorded).

## INTRODUCTION

Breast cancer is the most common cancer occurring in women, with 55 000 diagnoses and over 11 000 deaths in the UK annually.[1] Early detection strategies for the general population typically use mammography, with eligibility decided by age. However, there is significant debate around the balance of harms and benefits of age-based breast screening.[2–4] Furthermore, only around 30% of cancers are found by screening in the UK.[1] Offering the same breast screening strategy to the female population within a set age range does not take into account the fact that individual women may have very different breast cancer risks.[5–7] Clinical prediction models could be able to guide screening, prevention and treatment strategies, such as identifying those at higher risk to develop a breast cancer (overall, or a life-threatening tumour) or estimating prognosis after diagnosis is made.

'Risk-stratified breast screening' is a relatively novel concept, which suggests that targeting screening to those at highest risk might reduce the harms, and enhance the benefits of screening.[5] There may also be economic benefits to this approach.[8 9] Accurate, individualised risk prediction could also allow the identification of 'unrecognised' increased risk in other women and inform

preventive measures. While there are several risk prediction models available in this arena (table 1), a recent systematic review deemed that none are capable of guiding risk-stratified screening, due to model performance, and risk of bias concerns during model development and validation.[10] However, it is notable that this systematic review missed key papers, specifically the QCancer algorithms,[11] which to our knowledge represents the largest breast cancer risk prediction study undertaken, although this did not analyse model performance heterogeneity by time period, or relevant subpopulations. Indeed, it is increasingly advocated that while summary measures of model discrimination and calibration are clearly relevant, exploration of variability therein across clinically relevant subpopulations may present a more informative validation[12] as well as evaluation of transportability, especially when data are used across time periods (where there may be changes in treatments, diagnostics, incidence and mortality over time) and from different centres (where management pathways may differ).[13 14]

There has also been a recent increase in interest in the use of 'machine learning' modelling approaches to clinical prediction. However, there are concerns regarding the interpretability, transparency and robustness of validation for many of these models,[15] the fairness of methods to compare results from different modelling techniques,[15–17] and the extent to which often suggested better performance with machine learning is actually true.[18]

This study will develop and evaluate clinical prediction models for three endpoints: the risk of invasive breast cancer diagnosis, risk of developing and then dying from breast cancer, and the risk of dying from breast cancer after diagnosis. It will use the QResearch database comprising anonymised electronic health records data from over 1800 general practices in England to develop and robustly validate models for the prediction of breast cancer incidence and breast cancer death in women. Regression and 'machine learning' approaches to risk model development will be explored to perform a comparative evaluation of the utility of different techniques. Through an internal-external cross-validation (IECV) strategy,[19] comparisons with existing prediction models,[11] and external validation with an emphasis on investigating performance heterogeneity, this study seeks to develop highly performing prediction models that may be useful to guide risk-based breast cancer screening/care in the UK.

## METHODS AND ANALYSIS
### Study population and data sources for model development
An open cohort of women aged between 20 and 90 years at entry into the QResearch database will be identified (study years 2000–2020). Cohort entry will be the earliest of 20th birthday, or date of registration with the general practice plus 1 year. Study participants must have a recorded NHS number in the QResearch database to facilitate data linkages and be alive at the start of the study. Women with a pre-existing history of breast cancer or breast carcinoma in situ will not be eligible to enter the cohort.

The extracted QResearch cohort will be evaluated by descriptive statistics of key participant characteristics at cohort entry. Ascertainment of breast cancer cases and breast cancer deaths will be assessed by comparing the 'yields' from each of the four linked data sources by crude and age-standardised incidence rates. Crude and age-standardised incidence rates of incident breast cancer diagnoses and breast cancer mortality will be calculated for the study period overall, and for two phases, phase 1 (1 January 2000 to 31 December 2009) and phase 2 (1 January 2010 to 31 December 2020). The denominator will be the total of women included in the dataset/subset. Age-standardised rates will be calculated using direct standardisation based on 5-year age bands. Clinicopathological characteristics obtained from linked cancer registry data will be tabulated for breast cancer cases, and temporal trends in recording completeness will be examined.

### Outcome definitions
The three outcomes of interest are:
1. Developing breast cancer within the next 10 years.
2. Developing a breast cancer and then dying from it within the next 10 years.
3. Dying from breast cancer within 10 years of being diagnosed with breast cancer.

For the first outcome (breast cancer diagnosis), follow-up will be from cohort entry date to date of breast cancer diagnosis, or censoring (left cohort, reached end of study period alive or died from any cause). Leaving the cohort entails deregistering from the general practice. For this outcome, the competing risk of death from any cause will also be considered in modelling approaches. For the second outcome (breast cancer diagnosis and then death), women will be followed up from cohort entry until date of breast cancer death, or censoring (left cohort, reach study period end alive or die from a non-breast cancer cause). For this outcome, a competing risk of death from any cause other than breast cancer will be explored where appropriate. For outcome 3, (breast cancer mortality after diagnosis) follow-up will be from date of diagnosis to breast cancer death or censoring (leave the cohort, reach study end date alive or die from another cause). The competing risk for outcome three to also be investigated is death from a cause other than breast cancer.

For outcomes 1 and 2, models will be fitted using data from women without a previous history of breast cancer, or pre-cancerous conditions, for example, ductal carcinoma in situ. For outcome 3, models will only be developed using data from women with a confirmed incident diagnosis of invasive breast cancer.

For breast cancer diagnosis, this will be defined as the first date in which breast carcinoma is recorded in any

**Table 1** Comparison of selected key existing risk prediction models for developing breast cancer

| Risk model, or study first author (year) | Study setting | Parameters | Risk trajectory modelled | Validation strategy used | Discrimination metrics (95% CI) | Calibration metrics (95% CI) |
|---|---|---|---|---|---|---|
| Tyrer-Cuzick model, also known as 'IBIS' (2004)[49] | Model constructed from published data, informed by mathematical principles | Age, BRCA genotype, family history of breast cancer, (including relationship and ages), menarche, age at first birth, menopausal status, atypical hyperplasia, lobular carcinoma in situ, height, BMI | Diagnosis of breast cancer within next 10 years | None | Not reported | Not reported |
| van Veen (2018)[50] | UK-based cohort study, women aged 46–73 years attending screening centres (n=9363) | Tyrer-Cuzick model (TCM); TCM+mammographic density; TCM+mammographic density +PRS | Diagnosis of breast cancer within next 10 years | Predetermined scoring system applied to cohort | AUC 0.58 (0.52 to 0.62); AUC 0.64 (0.60 to 0.68); AUC 0.67 (0.62 to 0.71) | O/E 1.50 (1.33 to 1.70); O/E 0.98 (0.69 to 1.28) |
| 'Gail model' Gail, et al (1989)[51] | US-based case–control study, women aged 50+years (n=5998) | Age at menarche, age at first live birth, number of previous breast biopsies, number of first-degree relatives with breast cancer | Diagnosis of breast cancer within next 10, 20, and 30 years | None | Not reported | Not reported |
| Tice (2005)[52] | US-based cohort study, women aged 35+years (n=81 777) | Gail model; Gail model +breast density category | Diagnosis of breast cancer (study: median follow-up 5.1 years, no explicit horizon) | Apparent model performance | C-index 0.67 (0.65 to 0.68); C-index 0.68 (0.66 to 0.70) | None |
| 'BCSC model' Tice (2008)[53] | US cohort study of women aged 35+ years undergoing mammography (n=1 095 484) | Age, ethnicity, family history of breast cancer, breast biopsy history, breast density category | Diagnosis of breast cancer within next 5 years | Split-sample validation; Cross-validation | C-index 0.66 (0.65 to 0.67); C-index 0.66 (0.65 to 0.66) | O/E 1.03 (0.99 to 1.06) |
| QCancer Breast, Hippisley-Cox (2015)[11] | England-based primary care open cohort, women aged 25–84 years (n=3 318 258) | Age, BMI, deprivation, ethnicity, alcohol intake, family history of breast cancer, benign breast disease, OCP use, oestrogen-containing HRT use, manic depression/schizophrenia, previous blood cancer, previous lung cancer, previous ovarian cancer | Diagnosis of breast cancer within next 10 years | Split-sample validation | C-index 0.761 (0.758 to 0.765) | Calibration plots by tenth of predicted risk |

AUC, area under the receiver operating curve; BMI, body mass index; HRT, hormone replacement therapy; OCP, oral contraceptive pill; O/E, observed to expected ratio; PRS, Polygenic Rsk Score.

of the primary care records, Hospital Episode Statistics, cancer registry or Office for National Statistics death registry data. Breast cancer mortality will be defined as a recorded instance of breast cancer on the death certificate, either as the primary or as an underlying cause of death.

### Candidate predictor variables

Table 2 describes the variables that will be considered for inclusion in the models as predictors, based on published evidence, data from preclinical/molecular studies, their hypothesised nature as potentially affecting risk (such as effects on hormone regulation), or potential effect on outcomes after diagnosis of a breast carcinoma (such as treatment used). The latest recorded measurement prior to/at cohort entry will be used in the modelling, and duration between entry and measurement reported descriptively. Diagnoses that will be used as predictor variables will be defined as either being recorded in primary care data (Read/SNOMED codes) or on hospital records (International Classification of Diseases; ICD-10 codes).

### Modelling and evaluation strategy

The QResearch study dataset will comprise data collected across two decades, from over 1800 general practices in England. As regions may differ in terms of their baseline incidence of the outcomes of interest, predictor variable distributions (or effects on risk predictions), a paramount aspect of model evaluation will be to assess performance heterogeneity. Given temporal trends in baseline incidence and predictor effects, the performance of any model may deteriorate over time, but it is complex to estimate how well a model will perform after it starts to be deployed prospectively. Therefore, our evaluation strategy for each model will use IECV frameworks to validate models developed using the entirety of the QResearch data and simultaneously assess both geographical and temporal transportability (figure 1).

For the IECV, women entering the cohort during the first decade will have their follow-up time truncated to end at 31 December 2009, so as to preserve the temporal split. After IECV, random-effects meta-analysis will pool together performance estimates obtained for each geographical 'unit', which will either be individual general practices, or geographical region (n=10: East Midlands, East of England, London, North East, North West, South Central, South East, South West, West Midlands, Yorkshire and Humber).

Models will be then be assessed in an independent dataset for an external validation, which will include exploration of performance heterogeneity by age groups, ethnicities and geographical region. Missing data will be handled using multiple imputation where appropriate.[20 21]

Five forms of model will be explored:
1. Cox proportional hazards models.
2. Competing risks models.
3. Random survival forests (RSF)
4. 'Deep learning' neural networks (DL).
5. Extreme gradient boosting (XGBoost).

Each modelling approach will be used to develop risk prediction models for each of the three outcomes. The competing risks approach was selected in addition to the Cox proportional hazards model, as the latter may overestimate predictions in the setting of other events impeding the event of interest occurring.[22] Random forests, DL models and XGBoost were selected due to emerging evidence suggesting good performance on large, multidimensional datasets, and their ability to model complex, non-linear relationships.[23–26] Of note, generic DL neural networks and XGBoost do not account for right censoring, but DeepSurv represents a Cox proportional hazards neural network,[27] and a Cox model adaptation of XGBoost exists, which we will consider using in this study. We will consider 'standard' RSF as well as variants that can model competing risks[24]—for example, if competing risks models out-perform Cox regression-based model, the latter may be selected. For the purposes of a simple comparison, and to provide information regarding the association between increasing model complexity and performance, an age-only model be fitted for each outcome of interest (eg, a Cox model with age modelled as a restricted cubic spline with five knots). The TRIPOD Statement will be adhered to during study reporting.[28]

### Sample size calculations

We used the methods of Riley *et al* using the 'pmsampsize' package in R to derive sample size calculations for the development of the regression-based models.[29] We used data from Cancer Research UK regarding age-standardised rates of breast cancer incidence and breast cancer mortality in the UK population.[1] We assumed a mean follow-up of 6 years based on a recent QResearch cohort study examining prostate cancer-related outcomes in men aged 40–75 at cohort entry.[30] For each calculation, we used 15% of the maximum permitted Cox-Snell R-squared.[29]

#### Outcome 1: breast cancer incidence

For a time-to-event prediction model with 100 predictor parameters, a Cox-Snell R-squared of 0.072 (15% of 0.48; maximum permissible in this context), an age-standardised annual breast cancer diagnosis rate of 0.01665 (166.5/100 000[1]) in the cohort, we require 11 994 individuals with 1199 outcome events; 11.98 events per predictor parameter.

#### Outcome 2: breast cancer death

For a time-to-event prediction model with 100 predictor parameters, a Cox-Snell R-squared of 0.0045 (15% of 0.03; maximum permissible in this context), an age-standardised annual breast cancer mortality rate of 0.000334 (33.4/100 000,[1] we require 199 500 individuals with 400 outcome events; 4 events per predictor parameter.

**Table 2**  Summary of candidate predictor variables that will be considered in this study

| Variable class | Variables (and functional form) |
|---|---|
| Demographic variables | Age (continuous variable)<br>Townsend deprivation score (continuous)<br>Ethnicity (categorical, as per Office for National Statistics Census classes; white British, white Irish, other white background, white and black Caribbean, white and Black African, white and Asian, other mixed race, Indian, Pakistani, Bangladeshi, other Asian background, Caribbean, black African, other black background, Chinese, other ethnic groups (including Arab)) |
| Lifestyle factors | Smoking status (categorical, and also continuous if no of cigarettes per day is available)<br>Body mass index (continuous)<br>Alcohol intake (categorical; also continuous units per day if available) |
| Comorbidities and medical history (all binary, unless otherwise specified) | Previous ovarian cancer<br>Previous uterine cancer<br>Previous endometrial cancer<br>Previous ovarian cancer<br>Previous lung cancer<br>Previous haematological cancer<br>Previous thyroid cancer<br>Hypertension<br>Ischaemic heart disease<br>Diabetes mellitus type 1<br>Diabetes mellitus type 2<br>Cirrhosis of the liver/chronic liver disease<br>Systemic lupus erythematosus<br>Psychosis (incuding schizophrenia, depression with psychosis)<br>Fibromatosis or fibrocystic disease<br>Polycystic ovarian syndrome<br>Endometriosis<br>Chronic kidney disease (ordinal categorical, stages 3–5 (end-stage renal failure))<br>Vasculitis<br>Previous breast biopsies |
| Family history | Recorded family history of gynaecological cancer<br>Recorded family of breast cancer |
| Medications (at least three prescriptions prior to cohort entry; binary categorical) | Antihypertensives<br>Antipsychotics (atypical and typical)<br>Tricyclic antidepressants<br>Selective serotonin reuptake inhibitors<br>Monoamine oxidase inhibitors<br>Hormone replacement therapy<br>Oral contraceptive therapy |
| Reproductive history | No of pregnancies (continuous or ordinal categorical)<br>Menopause (binary; defined as recorded diagnosis of menopause on general practice or Hospital Episodes Statistics records, recorded prescriptions of hormone replacement therapy, or age at 60 at entry) |
| Tumour characteristics (for diagnosed tumours) | Stage at diagnosis (ordinal categorical, I–IV)<br>Tumour grade (ordinal categorical)<br>Lymph node involvement (binary)<br>Number lymph nodes excised (continuous)<br>Oestrogen receptor status (binary)<br>Progesterone receptor status (binary)<br>HER2 receptor status (binary)<br>Route to diagnosis (eg, 2-week referral, emergency presentation, screen detected) |
| Treatment variables (for diagnosed tumours) | Use of surgery<br>Use of chemotherapy<br>Use of radiotherapy |

As demonstrated, some classes of variables will only be appropriate for inclusion on models for certain outcomes of interest, that is, risk of death following a diagnosis of invasive breast cancer.

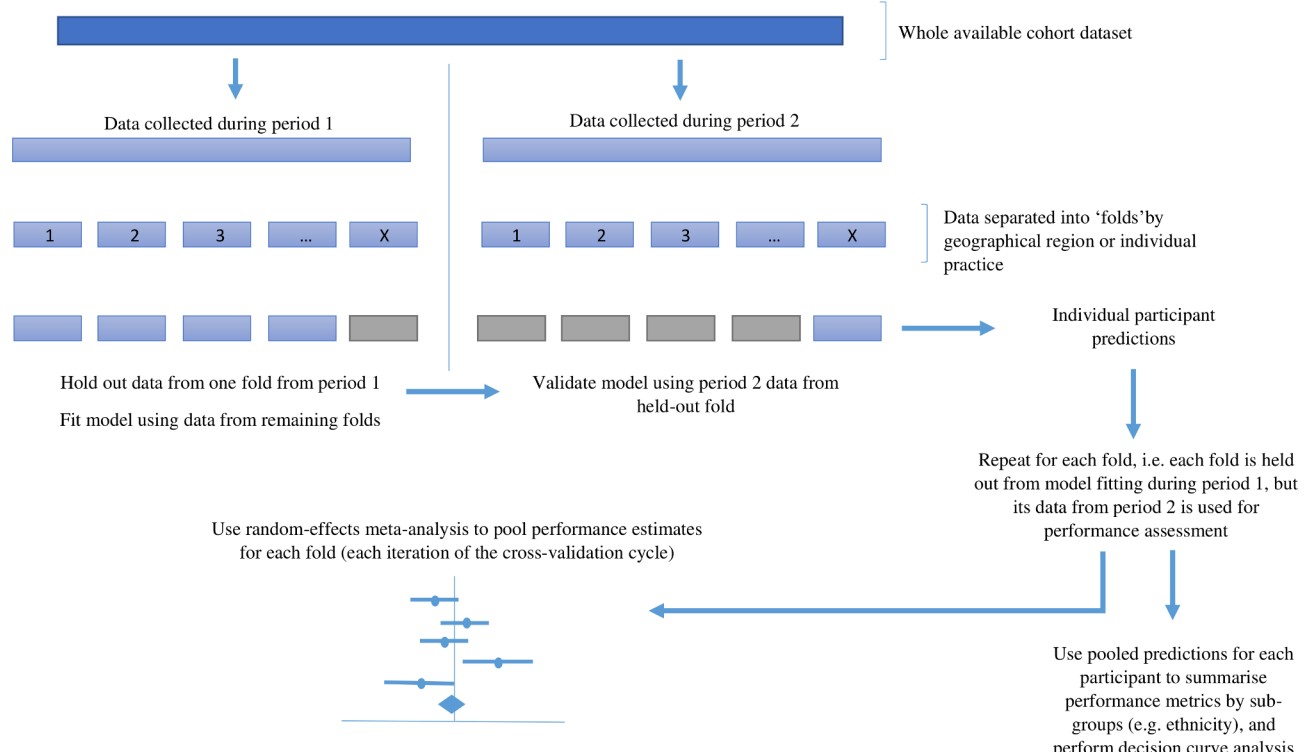

**Figure 1** Representation of the planned internal-external cross-validation schema that will concomitantly assess geographical and temporal transportability of each developed model. This permits the use of the entire dataset to develop and assess the performance of models, while also evaluating performance heterogeneity. Period 1 comprises 1 January 2000–31 December 2009; period 2 comprises 1 January 2010–31 December 2020.

## Outcome 3: breast cancer mortality following diagnosis

For a time-to-event prediction model with 100 parameters, and R-squared of 0.085 (15% of 0.57; maximum permissible in this context), an annual mortality rate of 0.024 (estimated from a 76% 10-year survival rate,[1] we require 10 080 individuals with 1452 outcome events; 14.52 events per predictor parameter.

Previous studies using earlier versions of the QResearch database within more restrictive time periods have identified cohorts of >4 million women[31]. QResearch contains records of over 100 000 cases of breast cancer, and preliminary evaluations suggest a final study cohort of over 11 million women.

There are no standard sample size calculations yet developed for risk prediction models using machine learning approaches. There is some evidence that machine learning models may require over 10 times the events per variable than regression-based methods[32]—even in this case, our planned study would have adequate sample size.

## Missing data handling

We anticipate missing data for ethnicity, body mass index (BMI), smoking status, alcohol intake and parity[33] (ie, older participants may not have parity recorded from births in their 20–30 s). For each of these variables, we will assess the extent and patterns of missingness, and consider multiple imputation with chained equations where the missing at random assumption is reasonable.[20] Distributions of variables will be assessed and if necessary,

appropriate transformations (eg, logarithmic) be used during the imputation process. In view of computational burden, five imputed data sets will be generated, and the imputation model will contain all candidate predictor variables and the outcomes of interest. The regression-based models will be developed and evaluated pooling over all five imputed datasets. As the parameters of RSF, DL and XGBoost models cannot be 'pooled' in accordance with Rubin's rules and they do not have SEs, these will be developed and evaluated using the first complete (imputed) dataset.

## Model development and evaluation: Cox and competing risks models

For the Cox and competing risks regression models for each outcome, continuous variables with non-linear effects on risk will be handled using restricted cubic splines, for example, age and BMI. We will include relevant interactions in the regression models, such as an interaction between age and family history of breast cancer. Full models will be fitted with all variables preselected by the researchers. The proportional hazards assumptions will be assessed using tests of Schoenfeld residuals.

Cox models and competing risks models will be developed on the entirety of each of the five imputed datasets with the model coefficients pooled in accordance with Rubin's rules to form the final models. Pooled model coefficients will be appropriately combined with the baseline survival function and cumulative incidence functions

at 10 years (respectively) to be able to make predictions. The 'apparent' performance of the models will be presented in terms of Harrell's C-index,[34] Royston's D statistic,[35] Brier score,[36] and calibration plots.[37] For competing risks models, the C index will be calculated using inverse probability of censoring weights.[38]

These models (which will be fitted to the entirety of the study data) will be validated using an IECV framework.[19 39] Herein, one practice will be held-out while the model is developed using data from all other practices during phase 1 (1 January 2000–31 December 2009). Harrell's C, Brier score, the calibration slope and calibration-in-the-large (CITL)[37 40] will be calculated using data from the held-out practice during phase 2 (1 January 2010–31 December 2020). The calibration slope refers to the 'spread' of estimated risks, where compared with an ideal value of 1, a slope<1 or>1 suggests that predictions are too extreme and too moderate, respectively.[37 40] CITL, also known as the calibration intercept where the ideal value is 0, may infer systematic overestimation or underestimation of true risk.[37 40]

This will be repeated for each individual practice (or, by region if computational expense is high). Random-effects meta-analysis will be used to pool the practice/region-level performance metric values with their standard errors to provide a summary estimate of model performance. The proportion of total variability in performance due to heterogeneity between practices/regions will be quantified by the $I^2$ (with 95% CI).[12] Prediction intervals (95%) will be calculated to estimate the spread of anticipated performance metrics of models in other datasets.[12] The pooled individual participant predictions will be used to assess performance in relevant subgroups, such by 10-year age bands and ethnic group. Therefore, the IECV approach allows assessment of overall model performance, model performance heterogeneity and model transportability.

Calibration curves will be generated for the held-out centres that have at least 100 events[41] and will be superimposed to demonstrate the 'spread' of calibration results. Point estimates for calibration slope, CITL and discrimination indices will be graphically displayed to depict performance heterogeneity, such as plotting values of Harrell's C-index by practice size, or practice-level estimates summarised by geographical region. Multivariable meta-regression will be used to examine potential contributory factors to heterogeneity (such as mean age of patients registered with each practice/in each centre, percentage non-white ethnicity groups, or mean deprivation score).[12 42–44]

The final models will be published in full, specifically the baseline survival (at 10 years) and all coefficients.

## Model development and evaluation: RSF, DL and XGBoost

For the DL and XGBoost models, categorical variables will be converted to dummy variables (often referred to as 'one hot encoding' in the machine learning literature). Continuous variables will be transformed using min-max scaling to constrain values between zero and one for the DL models.

These algorithms are often viewed as more flexible than the regression-based models as they may use 'hyperparameter tuning', wherein the parameters may be 'learnt' according to different settings, such as varying the number of trees in the random forest to find the highest performing arrangement, for example.[45 46] Using the whole dataset, 5-fold or 10-fold cross-validation will be used to identify optimal hyperparameters from a predetermined tuning grid and the model with the maximal c-statistic fitted to the entire data. As above, the apparent performance of each model will be presented (the averaged performance across cross-validation folds). The performance assessment of this final 'whole data' model will be performed by using the IECV framework in which the entire modelling strategy is replicated.

By using IECV, one practice will be 'held out,' with the data for remaining practices during phase 1 used to train a model using 5-fold or 10-fold cross-validation over a predefined tuning grid to find the optimal performing model (maximising for the c-statistic). The tuning grid will be the same as for the 'whole data' model. This model will then be tested in the data from the 'held out' practice during phase 2, with the c-statistic and its SE calculated, and the individual predictions stored. This will be repeated for every practice. The Brier score, calibration slope and CITL will also be calculated for each sequentially held-out practice. The final model's performance will be quantified using random-effects meta-analysis to pool the performance metric values with their standard errors, as above. Again, heterogeneity in performance due to between practice variation will be quantified by the $I^2$ (with 95% CI). Prediction intervals (95%) will be calculated to estimate the spread of anticipated performance metrics of models in other datasets. The pooled individual participant predictions will be used to display performance in relevant subgroups, such as ethnic groups.

Calibration curves will be generated for the held-out centres that have at least 100 events and will be superimposed to demonstrate the 'spread' of calibration results. Pooled model performance metrics and the heterogeneity ($I^2$ values) will be compared across model types. As for the regression-based models, multivariable meta-regression will be used to examine potential contributory factors to heterogeneity.

There are some concerns with the interpretability of machine learning models, which is at least partly attributable to their sometimes complex forms/structures. We will explore 'variable importance' approaches to estimate the effect that individual predictor variables have on model performance, and their contributions to generated risk predictions.

We will consider using an IECV strategy based on geographical regions (10) within the UK, rather than individual practices (>1800) if computational demand for RSF, DL and XGBoost models is extremely high. The final

models will be described in terms of relevant parameters, such as number of hidden layers, nodes and activation functions for DL.

## Decision curve analysis

As net benefit metrics may be useful to assess the clinical utility of a model, and also compare different forms of model, we will use decision curve analysis[47 48] to compare each model against standard 'age-based screening'.

## External validation

We will externally validate each model in an independent dataset comprising UK-based individuals, such as individuals from the UK Biobank cohort that were registered with general practices that do not contribute to QResearch. The UK Biobank is a prospective cohort of over 500 000 individuals, which underwent baseline assessments and phenotyping, with extant linkages to primary care, hospitalisation, cancer registry and death registry datasets. Based on the primary care, data fields in the UK Biobank, identification of the clinical software used by the participant's general practice is possible (ie, EMIS [Egton Medical Information Systems], or TPP [The Phoenix Partnership]). As the QResearch database is only linked to EMIS, this will present a method to identify individuals not in the primary dataset used for model development.

To validate models 1 and 2, women with a previous recorded diagnosis of breast cancer will be excluded, and follow-up will commence from date of UK Biobank assessment centre attendance, until the outcome of interest, death from another cause, date of withdrawal from UK Biobank, or right censoring (alive at last data extract). The external validation cohort will be evaluated by descriptive statistics of key participant characteristics. As for the development QResearch cohort, both crude and age-standardised incidence rates of breast cancer incidence and mortality will be calculated for the study period overall, and for 2000–2009 and 2010–2020 (if applicable) to consider any differences in endpoint incidence between development and validation cohorts.

Data for predictor variables will be handled and transformed identically as in the development/IECV stages. Multiple imputation with chained equations will be used to impute missing values for ethnicity, BMI, smoking status, alcohol intake, and parity (if appropriate) and will incorporate all predictor variables but exclude the outcome indicator. Five imputations will be generated.

Predictions for each participant in the UK Biobank cohort will be calculated for each model, in each imputed dataset. Performance metrics will be calculated in each imputed dataset and combined in accordance with Rubin's rules. An overall estimate of performance (Harrell's C-index, Royston's D statistic, Brier score, calibration slope, CITL) will be provided for each model along with its CI. Individual predictions will be combined across imputed datasets and used to derive calibration plots. Thereafter, the aforementioned metrics will be

calculated in sub-groups, that is, by separate geographical region, ethnic group and 10-year age group to demonstrate performance heterogeneity in clinically relevant subpopulations. For each model, performance metrics in the two data sources will be narratively compared, including the degree of heterogeneity.

## Statistical software

Multiple imputation, development, validation and meta-regression of Cox and competing risks models will be carried out in Stata V.17. The development, validation and meta-regression of the RSF, DL and XGBoost models will use R. Any commands or packages used will be reported in any manuscripts submitted for publication.

## Patient and public involvement

An institutional patient/public involvement network was used to identify volunteers (women affected by breast cancer) to feedback on the clinical need, research questions and study design. Qualitative feedback on the planned study was obtained from 'focus groups' of breast cancer support charities in the Oxfordshire region. Two patient/public involvement volunteers will be involved in the dissemination of results, such as being coauthors on publications if appropriate, and advising on the development of other communication methods.

## Ethics and dissemination

This project (reference OX129) has been approved by the QResearch scientific committee. The QResearch database annually obtains ethical approval from the East Midlands-Derby Research Ethics Committee (REC reference 18/EM/0400).

**Acknowledgements** We acknowledge the contribution of EMIS practices who contribute to QResearch and EMIS Health and the Universities of Nottingham and Oxford for expertise in establishing, developing or supporting the QResearch database. This project will involve data derived from patient-level information collected by the NHS, as part of the care and support of cancer patients. This data is collated, maintained and quality assured by the National Cancer Registration and Analysis Service, which is part of Public Health England (PHE). Access to the QResearch data is facilitated by the PHE Office for Data Release. The Hospital Episode Statistics data to be used in this analysis will be re-used by permission from NHS Digital who retain the copyright in that data. We thank in advance the Office of National Statistics for providing the mortality data. NHS Digital, Public Health England and Office of National Statistics bears no responsibility for the analysis or interpretation of the data provided.

**Contributors** Conceptualisation of the study: AKC, JH-C and GC. First draft of manuscript: AKC. Clinical input to study design: JH-C, DD and SL. dvanced statistical input to study design: GC and JH-C. Critical revision of manuscript: DD, SL, MB, SP, JH-C and GC

**Funding** AKC is funded by a Clinical Research Training Fellowship from Cancer Research UK (C2195/A31310 [Award: DCS-CRUK-CRTF20-AC], which is funding this project. JH-C reports grant to support the QResearch infrastructure including from National Institute for Health Research Biomedical Research Centre, Oxford, John Fell Oxford University Press Research Fund, Cancer Research UK (CR-UK) grant number C5255/A18085, through the Cancer Research UK Oxford Centre and Oxford Wellcome Institutional Strategic Support Fund (204826/Z/16/Z. DD reports funding from Cancer Research UK (C8225/A21133). GC reports a joint grant from Cancer Research UK and the NIHR Biomedical Research Centre, Oxford (C49297/A27294).

**Competing interests** JH-C is an unpaid director of QResearch (a not-for-profit organisation which is a partnership between the University of Oxford and EMIS

Health who supply the QResearch database) and is a founder and shareholder of ClinRisk and was its medical director until 31 May 2019 (ClinRisk produces open and closed source software to implement clinical risk algorithms (including a breast cancer prediction algorithm) into clinical computer systems.

**Patient and public involvement** Patients and/or the public were involved in the design, or conduct, or reporting, or dissemination plans of this research. Refer to the Methods section for further details.

**Patient consent for publication** Not applicable.

**Provenance and peer review** Not commissioned; externally peer reviewed.

**ORCID iDs**
Ashley Kieran Clift http://orcid.org/0000-0002-0061-979X
Julia Hippisley-Cox http://orcid.org/0000-0002-2479-7283
David Dodwell http://orcid.org/0000-0001-8787-4904
Stavros Petrou http://orcid.org/0000-0003-3121-6050
Gary S. Collins http://orcid.org/0000-0002-2772-2316

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
