## [Reviewer comments · BMJ Open]

ARTICLE DETAILS

TITLE (PROVISIONAL)	Development and validation of clinical prediction models for breast cancer incidence and mortality: a protocol for a dual cohort study
AUTHORS	Clift, Ashley; Hippisley-Cox, Julia; Dodwell, David; Lord, Simon; Brady, Mike; Petrou, Stavros; Collins, Gary

VERSION 1 – REVIEW

REVIEWER	Zhang, Chao Tianjin Medical University Cancer Institute and Hospital
REVIEW RETURNED	23-May-2021

GENERAL COMMENTS	The study was well designed. But it was written by a new hand. First, the authors should offer their results and conclusion sections in the manuscript. Second, seldom study referred table or figure in the introduction section. Third, future tense should not be widely used through the manuscript. I personally suggested the reject of the manuscript.
---

REVIEWER	Lemanska, Agnieszka University of Surrey
REVIEW RETURNED	01-Dec-2021

GENERAL COMMENTS	Thank you for the opportunity to review this study protocol. This is a very comprehensive and ambitious study. I look forward to reading the outputs in the future. I hope my comments are helpful in improving the clarity. Please mention in the introduction that both statistical and ML approaches will be used and discuss this. How will you tackle the interpretability aspect/issue? In terms of the external validation, please provide more information on the size of the UK Biobank primary care dataset and the availability of the linked data (Cancer Registration, HES and ONS) for the Biobank participants with primary care data. How do the authors know that the practices/participants that are in the QResearch are not included in the UK Biobank? The internal validation is not entirely clear either, the models will be developed using both periods with some of the centres held out for validation? What do the authors mean by a centre in Figure 1? GP practice or cancer centre/NHS Trust? The data from different Trusts may vary as the authors rightly say so how was this considered in deciding on the validation methodology? In the introduction authors appraise an important point about diagnosing and treating clinically relevant/not relevant cancers (overdiagnosis and overtreatment). This has however put me of the scent a little – how relevant is this for this study? Perhaps the introduction should refocus on the key aspects relevant to this
---

	study. To explain more, the authors are using cancer diagnosis as an outcome and not a diagnosis of clinically relevant cancers. Are they somehow treating “dying from cancer within 10 years” as a proxy for clinically relevant cancers? I would suggest not to muddle the waters, and focus your introduction around the important and relevant key points linked to the aims of the study to develop and validate models to predict 1, 2 and 3. This in itself is very important. In addition, please provide a reference to “preventive measures, such as chemoprevention” or rephrase. With regards to the exploration of temporal variability please explain more on why this is important in the context advancing clinical practice? So understandably, most probably there will be differences in the models and models' performance between data from 2000-10 and 2010-20. Why is the understanding of this difference important for improving clinical practice of today? It is important to use the most relevant/representative and up to date data in identifying risk factors and developing prediction models. In addition, regional variability is important so please specify what the region “unit” will be. The authors sate “geographical region” so does it mean counties or countries? Will NHS Trusts, commissioning units be compared? Which UK countries will be included and what is the coverage? For example what is the coverage of Cancer Registration across UK countries? The dual cohort design is complex. Please clarify on what the study entry is? Is it 2000 or a participant registration date with a practice (whichever is earlier?). What will this be for the two cohorts from different periods? Similarly, please explain what does leaving the cohort mean? How will the authors account for left censoring? In “must have a recorded NHS number” please add: recorded NHS number in QResearch database, because perhaps the participants may have their NHS number e.g. in Cancer Registration but not in the QResearch? Please give an estimated number of women without NHS number in QResearch that will be excluded? Is this at all possible not to have an NHS number assigned or extracted in primary care data? Will the inclusion criteria include the number of follow-up years for each participant? Some minor comments: The extracted QResearch cohort will be evaluated by descriptive statistics of key participant characteristics. – at which point in time? Study entry? Or at 10 year – study end/exit? Comparing the ‘yields’ from each of the 4 linked data sources – how will this be done? This is clearly explained in the paper later on, so perhaps this can be rephrased too or signposted. In: “incidence rates of breast cancer incidence and mortality” do you mean: rates of incident breast cancer and mortality? Or otherwise, this may need rephrasing. How will rates be calculated? Based on which denominator? Please explain this for both period1 and period2 cohorts and the overall cohort. Please change “oft” to often.
--	--

VERSION 1 – AUTHOR RESPONSE

Reviewer: 1

Dr. Chao Zhang, Tianjin Medical University Cancer Institute and Hospital

Comments to the Author:

The study was well designed. But it was written by a new hand. First, the authors should offer their results and conclusion sections in the manuscript. Second, seldom study referred table or figure in the introduction section. Third, future tense should not be widely used through the manuscript. I personally suggested the reject of the manuscript.

The Reviewer does not appear to recognise that this manuscript refers to a protocol, and therefore, the inclusion of results and conclusions would be inappropriate.

We have retained the use of tense due to personal style preference.

=====

Reviewer: 2

Dr. Agnieszka Lemanska, University of Surrey

Comments to the Author:

Thank you for the opportunity to review this study protocol. This is a very comprehensive and ambitious study. I look forward to reading the outputs in the future. I hope my comments are helpful in improving the clarity.

Thank you for your robust and helpful review.

Please mention in the introduction that both statistical and ML approaches will be used and discuss this.

This has been added.

How will you tackle the interpretability aspect/issue?

We have added some information on this (Model development and evaluation – RSF, NN and XGBoost, 5th paragraph) – briefly, we will explore techniques such as ‘variable importance’ to illustrate the relative contributions of predictor variables on the model performance/how the model generates predictions.

In terms of the external validation, please provide more information on the size of the UK Biobank primary care dataset and the availability of the linked data (Cancer Registration, HES and ONS) for the Biobank participants with primary care data. How do the authors know that the practices/participants that are in the QResearch are not included in the UK Biobank?

We have included further information on the above (Section: External validation) – such as the size of UK Biobank. Based on the primary care-related data fields in the UK Biobank, we are able to discern between practices that use EMIS or TPP software. As the QResearch database is linked only to the EMIS software system, we may be able to utilise data only for those with a TPP-using general practice.

The internal validation is not entirely clear either, the models will be developed using both periods with some of the centres held out for validation? What do the authors mean by a centre in Figure 1? GP practice or cancer centre/NHS Trust? The data from different Trusts may vary as the authors rightly say so how was this considered in deciding on the validation methodology?

Thank you pointing out this aspect for further clarification. The models will be developed using all available data (across the study period), then they will be validated using a form of cross-validation. Rather than randomly splitting the data into equally sized folds (such as with standard n-fold cross-validation), we seek to split the data by time period and region, as this permits a more robust assessment of model performance, specifically in terms of performance heterogeneity and transportability. The fact that different regions/practices may vary was a significant motivator for this, as our proposed IECV approach will permit us to assess the heterogeneity of model performance across these different areas.

We hope our edits to the section “Modelling and evaluation strategy” and Figure 1 makes this clearer for the Reviewer and future readers.

In the introduction authors appraise an important point about diagnosing and treating clinically relevant/not relevant cancers (overdiagnosis and overtreatment). This has however put me of the scent a little – how relevant is this for this study? Perhaps the introduction should refocus on the key aspects relevant to this study. To explain more, the authors are using cancer diagnosis as an outcome and not a diagnosis of clinically relevant cancers. Are they somehow treating “dying from cancer within 10 years” as a proxy for clinically relevant cancers? I would suggest not to muddle the waters, and focus your introduction around the important and relevant key points linked to the aims of the study to develop and validate models to predict 1, 2 and 3. This in itself is very important. In addition, please provide a reference to “preventive measures, such as chemoprevention” or rephrase.

Thank you for these useful suggestions. We have rephrased the first paragraph of the introduction accordingly, which we hope is more informative and helpful.

With regards to the exploration of temporal variability please explain more on why this is important in the context advancing clinical practice? So understandably, most probably there will be differences in the models and models' performance between data from 2000-10 and 2010-20. Why is the understanding of this difference important for improving clinical practice of today? It is important to use the most relevant/representative and up to date data in identifying risk factors and developing prediction models. In addition, regional variability is important so please specify what the region “unit” will be. The authors state “geographical region” so does it mean counties or countries? Will NHS Trusts, commissioning units be compared? Which UK countries will be included and what is the coverage? For example what is the coverage of Cancer Registration across UK countries?

As trends in incidence/mortality/predictor effects may change over time, it is complex to predict how well a model will perform after it starts to be implemented in a prospective setting. However, our intention is to use the IECV approach with the two decades of data to estimate how stable the model performances may be if they were eventually used in clinical practice. Therefore, the 2000-10 and 2010-20 split is not to understand prior temporal changes, but rather exploit the structure of the data to estimate the potential prospective performance of models. We have sought to clarify this in the manuscript, with edits to the section “Modelling and evaluation strategy”.

The units for geographical regions have been specified in the revised manuscript, and we have clarified that the dataset will only comprise England, for which the cancer registration is complete (i.e. 100% coverage of England).

The dual cohort design is complex. Please clarify on what the study entry is? Is it 2000 or a participant registration date with a practice (whichever is earlier?). What will this be for the two cohorts from different periods? Similarly, please explain what does leaving the cohort mean? How will the authors account for left censoring?

We have added information on these in the revised manuscript where relevant.

In “must have a recorded NHS number” please add: recorded NHS number in QResearch database, because perhaps the participants may have their NHS number e.g. in Cancer Registration but not in the QResearch? Please give an estimated number of women without NHS number in QResearch that will be excluded? Is this at all possible not to have an NHS number assigned or extracted in primary care data?

This clause has been added (“Study population and data sources for model development”). We are unable to estimate the number of women without an NHS number as this stage.

Will the inclusion criteria include the number of follow-up years for each participant?

Patients will need to be registered with their practice for at least 1 year before joining the cohort – this has been added to “Study population and data sources for model development.”

Some minor comments:

The extracted QResearch cohort will be evaluated by descriptive statistics of key participant characteristics. – at which point in time? Study entry? Or at 10 year – study end/exit?

Cohort entry – this has been added.

Comparing the ‘yields’ from each of the 4 linked data sources – how will this be done? This is clearly explained in the paper later on, so perhaps this can be rephrased too or signposted.

This has been rephrased.

In: “incidence rates of breast cancer incidence and mortality” do you mean: rates of incident breast cancer and mortality? Or otherwise, this may need rephrasing. How will rates be calculated? Based on which denominator? Please explain this for both period1 and period2 cohorts and the overall cohort.

This has been clarified in the revised manuscript.

Please change “oft” to often.

Changed.

VERSION 2 – REVIEW

REVIEWER	Lemanska, Agnieszka University of Surrey
REVIEW RETURNED	06-Jan-2022
GENERAL COMMENTS	A great manuscript and thank you for addressing the comments raised in the initial review. Good luck with your study!